# PSPC1 Potentiates IGF1R Expression to Augment Cell Adhesion and Motility

**DOI:** 10.3390/cells9061490

**Published:** 2020-06-18

**Authors:** Hsin-Wei Jen, De-Leung Gu, Yaw-Dong Lang, Yuh-Shan Jou

**Affiliations:** 1Graduate Institute of Life Sciences, National Defense Medical Center, Taipei 11490, Taiwan; hwjen@ibms.sinica.edu.tw; 2Institute of Biomedical Sciences, Academia Sinica, Taipei 11529, Taiwan; mito8333@gmail.com (D.-L.G.); langyd@ibms.sinica.edu.tw (Y.-D.L.)

**Keywords:** PSPC1, paraspeckle, IGF1R, NONO, FUS, lncRNA *Neat1*, FAK/Src, cell motility

## Abstract

Paraspeckle protein 1 (PSPC1) overexpression in cancers is known to be the pro-metastatic switch of tumor progression associated with poor prognosis of cancer patients. However, the detail molecular mechanisms to facilitate cancer cell migration remain elusive. Here, we conducted integrated analysis of human phospho-kinase antibody array, transcriptome analysis with RNA-seq, and proteomic analysis of protein pulldown to study the molecular detail of PSPC1-potentiated phenotypical transformation, adhesion, and motility in human hepatocellular carcinoma (HCC) cells. We found that PSPC1 overexpression re-assembles and augments stress fiber formations to promote recruitment of focal adhesion contacts at the protruding edge to facilitate cell migration. PSPC1 activated focal adhesion-associated kinases especially FAK/Src signaling to enhance cell adhesion and motility toward extracellular matrix (ECM). Integrated transcriptome and gene set enrichment analysis indicated that PSPC1 modulated receptor tyrosine kinase IGF1R involved in the focal adhesion pathway and induction of diverse integrins expression. Knockdown IGF1R expression and treatment of IGF1R inhibitor suppressed PSPC1-induced cell motility. Interestingly, knockdown PSPC1-interacted paraspeckle components including NONO, FUS, and the lncRNA *Neat1* abolished PSPC1-activated IGF1R expression. Together, PSPC1 overexpression induced focal adhesion formation and facilitated cell motility via activation of IGF1R signaling. PSPC1 overexpression in tumors could be a potential biomarker of target therapy with IGF1R inhibitor for improvement of HCC therapy.

## 1. Introduction

Interactions of cell-to-cell adhesion and cell-to-extracellular matrix (ECM) play critical roles in sustaining tissue homeostasis and microenvironment to augment cell motility toward their destinations during normal tissue ontogenesis and tumor progression, respectively. Normal epithelial cells along with other cell types build up and maintain the organ architecture by adhering to distinct proteins of adhesion and ECM to anchor cells and to trigger downstream physiological signaling for specific organ development. Cancer cells dominantly interact with stroma cells to orchestrate the microenvironment for malignant tumor progression.

Paraspeckles contain several well-characterized members of the Drosophila behavior human splicing (DBHS) family proteins including paraspeckle component 1 (PSPC1), non-POU domain- containing octamer binding protein (NONO), and splicing factor proline- and glutamine-rich (SFPQ). DBHS proteins shared consensus sequence containing domains of RNA recognition motifs (RRM), NONA/paraspeckle (NOPS) and coiled-coil. DBHS proteins are frequently identified to participate in every step of gene regulation including transcriptional regulation, post- transcriptional modulations of RNA stability, and RNA processing and transport [1,2,3]. PSPC1 is the first identified structural protein of the sub-nuclear foci within nuclear paraspeckle and is known to play a function in DNA damage response [4,5], adipocytes differentiation [6], viral gene regulation [7], and RNA epigenetic modulation and gene regulation [3]. By now, more than 40 proteins have been identified as paraspeckle-associated proteins, among SFPQ, NONO, FUS [1,8,9], and a long non-coding RNA (LncRNA) *Neat1* are essential components required for paraspeckle formation [10]. Recently, PSPC1 upregulation in multiple cancer types was demonstrated to play as a contextual determinant of pro-metastatic switch via hijacking the Smad2/3 from targeting pro-apoptotic genes in normal cells reprogrammed to activate TGF-β1 autocrine signaling and the pro-metastatic target genes in cancer cells to facilitate tumor progression [11]. PSPC1 is also a substrate of protein tyrosine kinase 6 (PTK6) but sequestered PTK6 in the nucleus and abolished the PSPC1 oncogenic functions in human hepatocellular carcinoma (HCC) cells [12].

Insulin-like growth factor 1 receptor (IGF1R) is a transmembrane receptor tyrosine kinase (RTK) frequently found to be upregulated and associated with cancer progression and patients’ poor prognosis in multiple cancer types including HCC [13,14,15,16,17]. Overexpression of IGF1R activates canonical downstream targets PI3K/AKT and MAPK/ERK signaling [18], that are critical for tumorigenic functions including cell growth, survival, migration, epithelial to mesenchymal transition (EMT), and drug resistance [19,20]. In addition, IGF1R synergies with cell surface receptor integrins for activation of non-canonical FAK and/or Src kinases to promote focal contact maturation and cytoskeleton remodeling [21,22,23]. Activation of IGF1R promotes cellular motility through altered cell surface integrin expression by activation of downstream IRS2, FAK, RHOA, ROCK signaling led to tumor invasion and metastasis [24,25].

Here, we provide lines of evidence that PSPC1 augments cell adhesion and motility via promoting IGF1R expression to stimulate downstream focal adhesion and integrin signaling pathways including integrin/FAK/Src and AKT axes. We also investigate the impacts of paraspeckle component proteins and their binding scaffold lncRNA *Neat1* participated in PSPC1/IGF1R axis-potentiated cell motility. Our results might provide molecular and mechanistic insights for the possibility of using the PSPC1/IGF1R oncogenic pathway for developing innovative theranostic biomarkers and therapeutic strategy.

## 2. Materials and Methods

### 2.1. Cell Culture and Constructs

Human HCC cell lines (SK-Hep1, PLC5, and Hep3B) were maintained in Dulbecco’s Modified Eagle Medium containing 10% fetal bovine serum and 1% penicillin/streptomycin [12]. Short hairpin RNAs (shRNA) targeting PSPC1 and IGF1R were purchased from RNAi core facility (National RNAi Core Facility, Academia Sinica, Taipei, Taiwan) listed in Appendix A. Human IGF1R in the pBabe-bleo retroviral vector purchased from Addgene (Clone ID: #11212), PSPC1 in the pcDNA3-HA (Addgene Clone ID: #101764), or PSPC1 shRNAs in the pLKO.1 lentivirus infection system was introduced into HCC cells individually and selected stable cells by using zeocine (100 µg/mL), neomycin (200 µg/mL), or puromycin (10 µg/mL) respectively for 2 weeks according to the standard protocols [11]. Mock indicated the control experiment with empty vector only and selected stable control cells with corresponding antibiotics.

### 2.2. Reagents

Corning® Matrigel® Growth Factor Reduced (GFR) Basement Membrane Matrix containing purified human collagen I, laminin 1, vitronectin, and fibronectin was purchased from CORNING, NY, USA. DNA constructs and expression vectors were transfected into targeting cells by using Turbofect transfection reagent (Thermo-Fisher Scientific, Waltham, MA, USA). RIPA lysis buffer 10× was obtained from EMD Millipore (#20-188). Tablets of the cOmplete™ EDTA-free Protease Inhibitor Cocktail and the PhosSTOP™ Phosphatase inhibitor were purchased from Roche. The proteome profiler human phospho-kinase array kit was purchased from R&D Systems (Minneapolis, MN, USA) (catalog # ARY003B).

### 2.3. Adhesion Assay

First, 96-well plates were coated with 50 µL per well of purified matrix proteins (10 µg/mL) ordered from Corning® (Collagen I: #354236, laminin: #354232, fibronectin: #356008 and vitronectin: #354238) overnight at 4 °C. The plates were PBS washed and blocked with 2% BSA/PBS for 2 h at 37 °C. Cells were then PBS washed and treated with trypsin-EDTA (Promega) to obtain single cell suspension at 37 °C. Cells were re-suspended in the DMEM at a concentration of 2 × 10^5^ cells/mL. Then, 100 µL of cells was added into each well to incubate for 30 min at 37 °C. The unattached cells were removed by decanting the plates followed by PBS washing three times. The attached cells were fixed with 4% paraformaldehyde, stained with 0.5% crystal violet, and counted in five randomly selected fields under 400× magnification.

### 2.4. 3D Matrigel Assay

The Matrigel (250 µL) was spread into wells on a 48-well plate. One thousand suspended cells were plated on the top of Matrigel-coated wells. After incubation at 37 °C for 20 min to allow the cells to attach on the top of the gel, medium was replaced with complete growth medium containing 5% Matrigel. Cells were cultivated and medium was changed with fresh medium containing Matrigel every two days. Images of cells were taken by microscopy.

### 2.5. Cell Lysate and Immunoblot Analysis

Cell lysates were prepared in RIPA lysis buffer containing protease inhibitor and phosphatase inhibitor. Protein samples were separated by SDS-PAGE and then transferred to polyvinylidene difluoride (PVDF) membrane and followed by Western blotting protocol treated with primary antibody to recognize the target antigen. The antibodies in this study are listed in Appendix A.

### 2.6. Immunofluorescence Analysis

Cells were fixed in 4% paraformaldehyde at room temperature for 15 min, blocked with blocking buffer (2% BSA and 0.2% Triton X-100 in PBS), and then incubated with primary antibody for 45 min. Nuclei were counterstained with DAPI (Sigma) for 2 min.

### 2.7. Migration, Invasion and Haptotaxis Assays 

For the invasion assay, the cell culture inserts (8 µm PET, Millicell Cell Culture Inserts) were evenly coated with diluted Matrigel (1:5 dilution within total 50 µL serum-free medium). Cells (1 × 10^4^ to 5 × 10^5^) were added to the upper chamber and the lower chamber was filled with 700 µL medium containing 10% FBS maintained for 24 h. The cell migration assay was similar to the invasion assay, except that inserts were not coated with Matrigel. For the haptotaxis assay, the cell culture inserts were coated on the underside with purified matrix proteins (10 µg/mL in PBS) at 4 °C overnight followed blocking with 2% BSA in PBS at 37 °C for 2 h. Cells (1 × 10^5^) in serum-free medium were added to the upper chamber. The lower chamber was filled with 700 µL of the same medium maintained for 3 h. Cells were then fixed with 4% paraformaldehyde for 15 min and stained with 0.5% crystal violet for 10 min. The cells on the upper side of the filters were removed with cotton-tipped swabs. All the cells on the underside of the filters were counted.

### 2.8. Transcriptome Analysis and Proteomic Analysis

To elucidate the downstream pathways modulated by PSPC1 expression in transcriptome analysis, firstly, we determined the consensus differential expression genes (DEGs, ±1.5-fold change) shared in between DEGs of PSPC1 over-expressing (SK-Hep1) and PSPC1 knockdown (Hep3B) HCC cells versus their corresponding mock cells. We obtained 684 PSPC1-modulated genes after overlapping 3405 upregulated and 3921 downregulated DEGs. Secondly, to demonstrate enriched pathways in 684 PSPC1-modulated genes, we conducted pathway enrichment analysis through Gene Set Enrichment Analysis (GSEA) and Gene Set Analysis Toolkit (GSAT), and displayed the top 10 enrichment pathways by histogram graphs based on pathway information from KEGG, Pathway commons, and Wikipathways data set. Red bars indicated the enriched cell adhesion and migration-related pathways based on their collections. Finally, to determine the common PSPC1-modulated genes shared in these cell adhesion and migration-related pathways (red bars), we illustrated DEGs in the pathways with heatmaps and identified seven overlapping genes among independent pathway analysis shown as a Venn diagram. For proteomic analysis, the proteomic datasets and analysis were based on previous reports followed by STRING analysis to show PSPC1-interacting networks [12,26].

### 2.9. RRM Domain Mutation and Deletion Construction of PSPC1

We generated the PSPC1 RRM mutation (PSPC1-RRMmut) construct with four point mutations in RRM1 (F118A, F120A) and RRM2 (K197A, F199A) domains, and the PSPC1 RRM deletion (PSPC1-ΔRRM) construct with deletions of RRM1 and RRM2 domains. Both constructs were subcloned into pcDNA3.0-Flag vector [11].

## 3. Results

### 3.1. Expression of PSPC1 Promotes Cell Migration and Cytoskeleton Re-Assembly

To explore the roles of PSPC1 in cell motility, we selected three human hepatocellular carcinoma (HCC) cell lines including SK-Hep1 (PSPC1 deficient due to homozygous deletion) and PLC5 (low PSPC1 expression) cells for PSPC1 overexpression and Hep3B (high PSPC1 expression) cells for PSPC1 knockdown in the functional assays with validation of PSPC1 expression by Western blotting (Figure 1A). Consistent with previous reports [11,12], overexpression of PSPC1 substantially increased cell migration and invasion ability in SK-Hep1 and PLC5 cells based on the Boyden chamber assays (Figure 1B,C). Conversely, PSPC1 knockdown by shRNAs (*shPSPC1 #9* and *#10*) in Hep3B cells diminished cell migration and invasion as compared to the control group (Figure 1D). Moreover, we found that overexpression of PSPC1 in SK-Hep1 cells showed extended invasive branching cellular protrusions compared with that of mock control cells in Matrigel-embedded 3D cultured condition (Figure 1E). Since elevation of cell motility is generally accompanied with phenotypical transformation including cell spreading, cytoskeleton remodeling, polarized protrusion, and focal adhesion formation [27], we found that expression of PSPC1 obviously induced cell morphology transformation in SK-Hep1 and PLC5 that exhibited augmented cell spreading phenotype compared with mock cells (Figure 1F). To detect whether polymerization of actin filaments was induced by PSPC1, phalloidin staining was conducted with PSPC1 expressing SK-Hep1 and PLC5 cells, and found that PSPC1 increased actin filaments assembly extension from focal adhesion complex protein Talin1/2 (Figure 1G). Finally, by using phalloidin labeling assay for detecting the distribution of filamentous actin (F-actin) stress fiber [28], we found that PSPC1 significantly augments protrusive structure filopodia and sheet-like lamellipodia in SK-Hep1 and PLC5 cells, respectively (Figure 1H). Together, we concluded that overexpression of PSPC1 in HCC cells augments F-actin stress fiber formations to promote recruitment of focal adhesion contacts at the protruding edge to induce morphological transformation with a spreading phenotype to facilitate cell migration.

### 3.2. PSPC1 Induced Focal Adhesion Formation and Chemotaxis to ECM

To uncover the underlying signal transduction pathways that participated in PSPC1-augmented cell migration, we collected whole cell lysates of PSPC1-overexpressing SK-Hep1 cells and detected activated kinases in phospho-kinase array. Our results indicated that overexpression of PSPC1 activated several kinases activity including FAK, Src, Akt, and Chk-2, but suppressed phosphorylation in AMPKα1, p38α, and p53 (Figure 2A). Since FAK/Src signaling are known to participate in focal adhesion complex formation and cytoskeletal reorganization, we conducted Western blotting analysis of p-FAK(Y397), p-FAK(Y576/577), p-Src(Y416) and their total protein forms for confirming the effect of PSPC1 in focal adhesion-associated kinases. Our results showed that PSPC1 robustly increased FAK and/or Src phosphorylation in SK-Hep1 and PLC5 cells (Figure 2B). In contrast, PSPC1 depletion significantly decreases both kinase activities in Hep3B cells (Figure 2B). To further confirm the effect of PSPC1 in focal adhesion formation, immunofluorescent staining of two components of focal adhesion complexes paxillin and Talin1/2 was investigated in PSPC1-overexpressing HCC cells. The results showed the expression of paxillin phosphorylation (p-Y118) colocalized with Talin1/2 in SK-Hep1 and PLC5 cells at the nascent adhesions of the protruding cell edges (Figure 2C). Together, we concluded that overexpression of PSPC1 in HCC cells assembles F-actin stress fibers formations derived from focal adhesion contacts at the protruding edge to facilitate cell migration. Since focal contacts formation is required for dynamic interactions of mobile cells with extracellular matrix (ECM) for cell mobility, we then evaluated whether PSPC1 enhances cell adhesion and migration via increasing responses to common ECM proteins including type I collagen (COL I), laminin (LN), fibronectin (FN), and vitronectin (VN) by their attachment capabilities. Overexpression of PSPC1 in SK-Hep1 and PLC5 cells significantly enhanced cell adhesion to collagen, laminin, and fibronectin (Figure 2D), as well as increased cell migration on collagen and laminin (Figure 2E). Conversely, PSPC1 knockdown by shRNA significantly impaired cell migration on the type I collagen, laminin, and fibronectin, but had less impact on cell adhesion ability in Hep3B cells (Figure 2D,E). In summary, our results indicated that overexpression of PSPC1 activated focal adhesion kinases FAK/Src in HCC cells to augment phosphor-paxillin colocalized with Talin1/2 at the nascent adhesions of the protruding cell edges. The PSPC1-activated focal adhesion-associated signaling leads to enhanced F-actin stress fiber formations on the protruding cell edges toward diverse ECM proteins on different HCC cells to facilitate cell motility.

### 3.3. PSPC1 Modulates IGF1R Expression Involved in Focal Adhesion Pathway

To explore molecular details of PSPC1-mediated signaling pathways related to cellular focal adhesion and motility, we performed RNA-seq transcriptome analysis for detection of PSPC1- modulated genes derived from PSPC1 overexpression in SK-Hep1 cells (GSE114856) and PSPC1 knockdown in Hep3B cells (GSE148551). A total of 684 DEGs (differential expression genes) were identified by selection of the common altered genes shared in between DEGs of PSPC1 overexpression cells (fold change cutoff ≥ 1.5, *n* = 3405) and PSPC1 knockdown cells (fold change cutoff ≤ 1.5, *n* = 3921) (Figure 3A). Consistent with the experimental results, the focal adhesion pathway is listed on the top signature of the KEGG gene set enrichment analysis of 684 common DEGs (GSEA, *p*-value = 2.53x10^-8^) (Figure 3B). Furthermore, two integrin-ECM interactions associated pathways (integrin family cell surface interactions, *p*-value = 3.61 × 10−^10^; Beta1 integrin family cell surface interactions, *p*-value = 1.22 × 10^−9^) were listed as the top candidates in analysis of pathway common gene sets (Figure 3C). The focal adhesion pathway (*p*-value = 0.0004) was also ranked as a significant signature of Wikipathway by Gene SeT AnaLysis Toolkit (GSAT) analysis (Figure 3D).

To identify the key PSPC1-modulated genes that participated in the gene signatures of integrin and focal adhesion pathways, we displayed the genes listed in the three gene signatures and showed the intensities of altered gene expression in heat maps (Figure 3E). We identified seven genes that were in common in the three gene sets including LAMB1, LAMA5, COL1A2, COL5A2, ITGA10, IGF1R, and PDGFRβ that were further selected for expression and functional validations (Figure 3F). Among them, only IGF1R displays a consistent regulatory pattern after PSPC1 expression or knockdown in three HCC cells at mRNA (Figure 3G–I) and protein levels (Figure 3J). To evaluate whether PSPC1 regulates IGF1R expression widespread in other HCC cells, we knocked down PSPC1 in three additional HCC cell lines Huh7, HepG2, and Mahlavu cells and showed consistent consequence in downregulation of IGF1R expression (Appendix A). 

To investigate whether PSPC1 modulates IGF1R downstream canonical signaling AKT and ERK, we found that p-AKT (S473) and p-ERK (T202/Y204) were activated in SK-Hep1 cell, whereas depletion of PSPC1 suppressed the AKT and ERK phosphorylation in Hep3B cells (Figure 3K). As PSPC1 stimulates cell adhesion and migration through ECM with enriched pathways associated with integrin cell surface interactions, we investigated whether PSPC1 modulates integrin expression that responds to type I collage and laminin via PSPC1 expression or knockdown in three HCC cells. Our results showed that PSPC1 robustly increased integrin β4 and β1 expression in SK-Hep1 and PLC5 cells, respectively, whereas knockdown of PSPC1 decreased α2, α6, and β1 integrins expression in Hep3B cells (Appendix A). These results suggest that PSPC1 diversely regulated different integrins expression via an unknown intermediate in different HCC cells. Collectively, our results indicated that PSPC1 is a novel regulator of IGF1R that involved in activation of PSPC1-medidtaed focal adhesion pathways to facilitate cell migration. IGF1R is a downstream target of PSPC1 involved in focal adhesion pathway.

### 3.4. Blocking IGF1R Signaling Suppresses PSPC1-Induced Cell Motility and Adhesion

To investigate the impacts of IGF1R in PSPC1-enhanced cell adhesion and migration, we specifically silenced IGF1R by shRNA (shIGF1R #31 or #35) in PSPC1-overexpressing HCC cells (Appendix A). We found that IGF1R depletion eliminated the PSPC1-induced cell migration and adhesion to type I collagen in SK-Hep1 and PLC5 cells (Figure 4A,B). Furthermore, blocking IGF1R also suppressed PSPC1-induced invasiveness in three-dimensional culture (upper) and stress fiber assembly (bottom) (Figure 4C). IGF1R was reported to enhance stability of β1 integrin at protein levels by interacting with cytoplasmic domain and co-localized in cell adhesion complexes [23,29,30], disrupted complex formation of IGF1R and integrin β1 could suppress the cooperative signaling and cytoskeleton organization and migration [31]. In addition, IGF1R silencing decreased integrin β4 expression and suppressed cell adhesion capacity in NHSCC cells [32]. For the impacts to the IGF1R downstream integrin/FAK/Src signaling, IGF1R knockdown markedly inhibited PSPC1-induced integrin β4 and β1 expression in SK-Hep1 and PLC5 cells, respectively, as well as suppressed FAK and/or Src activation in both HCC cells (Figure 4D,E). 

To evaluate whether restoration of IGF1R expression could rescue cell migration and adhesion in PSPC1 knockdown cells, we exogenously expressed IGF1R in PSPC1-knocked down Hep3B cells and found that IGF1R restored cell motility, but failed to rescue cell adhesion capacity (Appendix A). In addition, we found that restoration of IGF1R recovered the p-AKT and p-ERK activity without affecting PSPC1 expression (Appendix A), while unable to restore FAK and Src phosphorylation (Appendix A). These results suggested that PSPC1 induced IGF1R expression as well as activated crosstalk signaling between FAK/Src and AKT/ERK involved in PSPC1-induced cell motility.

To further validate the IGF1R signaling in PSPC1-potentiated cell motility, we treated the PSPC1-overexpressing HCC cells with IGF1R specific inhibitor cyclolignan picropodophyllin (PPP) a potent inhibitor blocked its expression and activity without affecting other insulin family receptors [33,34]. Our results indicated that PPP treatments of HCC cells significantly decreased PSPC1-induced cell migration either ectopic or endogenous expression of PSPC1 (Figure 4F–H) as well as FAK/Src and AKT kinase activities in a dose-dependent manner (Figure 4I–K). In summary, our results indicated that PSPC1 elevated IGF1R expression led to activation of FAK/Src and AKT signaling that promotes cell motility in HCC cells.

### 3.5. PSPC1 RNA Recognition Motif (RRM) Required for PSPC1/IGF1R Axis Activated Cell Motility

Since DBHS protein domains NOPS and c-terminal coiled-coil domain were considered exclusively in mediating DBHS dimerization and oligomerization [1], we speculated that the remaining RNA recognition motifs (RRMs) might play critical roles in PSPC1/IGF1R axis modulating PSPC1-augmented cell migration. We constructed PSPC1 RRM1 and RRM2 mutant (PSPC1-RRMmut: F118A and F120A in RRM1 and K197A and F199A in RRM2) [35,36] and deletion (PSPC1-ΔRRM) mutant of PSPC1 (Figure 5A) to delineate their involvement in PSPC1-augmented cell migration in SK-Hep1 and PLC5 cells. Our results indicated that PSPC1-RRMmut and PSPC1-ΔRRM failed to increase expression of IGF1R at mRNA (Figure 5B,C) and protein levels (Figure 5D,E) as compared with that of wild type PSPC1. In addition, PSPC1-induced cell invasion capacity was also impeded in PSPC1-RRMmut and PSPC1-ΔRRM stably expressed HCC cells compared to that of wild type (Figure 5F,G). We further assessed the effect of both mutants on integrin modulation, and found that PSPC1 mutants did not elevate integrin β4 and β1 expression in SK-Hep1 and PLC5 cells, respectively (Figure 5H,I). These findings suggest that PSPC1 RRM1 and RRM2 domains are required for IGF1R induction as well as affecting PSPC1-enhanced cell motility.

### 3.6. PSPC1 Collaborated with Paraspeckle Components to Modulate IGF1R Expression

Since PSPC1 is one of the core proteins of paraspeckles that are highly connected with many essential components including NONO, FUS, other paraspeckle-associated proteins and the scaffold long non-coding RNA (lncRNA) *Neat1* for sustaining the paraspeckle structure, we speculated that PSPC1-binding partners might collaborate with PSPC1 for modulation of IGF1R expression. We conducted PSPC1 specific antibody pulled down, separated proteins by SDS-PAGE, and sliced interacting proteins bands, performed mass spectroscopy analysis, and identified 141 PSPC1 binding proteins (Appendix A). Interestingly, we found those PSPC1-interacting proteins are highly concordant with the known paraspeckle proteins as reported by previous studies [37,38]. We ranked the PSPC1-interacting proteins based on the score of mass spectrum, conducted protein–protein association networks analysis and revealed the PSPC1-networking proteins including FUS, NONO, SFPQ, RBM14, HNRNPK, HNRNPM by using STRING network analysis (Figure 6A,B) [26]. Based on the top-ranking MS spectrum score and PSPC1 protein networking, we selected FUS and NONO (blue bars) with knockdown analysis for examining their involvement in PSPC1-enhanced IGF1R expression. We found that depletion of FUS and NONO expression by using their corresponding small interfering RNAs (siRNAs) in PSPC1-overexpressing SK-Hep1 and PLC5 cells significantly suppressed PSPC1-induced IGF1R expression (Figure 6C,D). To confirm FUS and NONO are necessary for IGF1R induction, we silenced FUS and NONO in parental HCC cells of SK-Hep1, Hep3B, and Huh7 without additional expression of PSPC1 and showed the same reduction of IGF1R expression (Figure 6E,F). Moreover, we also examined the role of scaffold lncRNA *Neat1* of paraspeckle for the impact of PSPC1-modulated IGF1R expression. We knocked down *Neat1* using siRNA in four HCC cell lines SK-Hep1, PLC5, Hep3B, and Huh7 and showed a consistent decrease of IGF1R expression pattern like the blocking expression of FUS and NONO (Figure 6G). Collectively, our results suggested that paraspeckle components are required for PSPC1-mediated IGF1R expression and potentially participated in PSPC1/IGF1R axis-enhanced cell motility.

## 4. Discussion

Recently, we reported that PSPC1 could potentiate TGF-β1 autocrine signaling to cancer cells and function as the master activator of epithelial to mesenchymal transition (EMT) and the stemness via activation of EMT and stemness core transcription factors such as Snail, Slug, Twist and Nanog, Oct4, Sox2, respectively [39,40]. Meanwhile, when PSPC1 is overexpressed in HCC cells, PSPC1 not only loses its sequestration of tumor suppressive PTK6 in the nucleus but also facilitates PTK6 cytoplasmic translocation to be an oncogene and β-catenin nuclear translocation to interact with PSPC1 for augmenting Wnt3a autocrine signaling and tumor progression [41]. In this study, we provide the molecular details of PSPC1-potentiated cell adhesion and motility via IGF1R signaling axis with approaches of transcriptome analysis to identify the consensus IGF1R upregulation and the phospho-kinase array to identify the activation of FAK/Src and canonical AKT signaling. Moreover, with mutations of RRM motifs of PSPC1 to abolish activation of IGF1R gene expression, we demonstrated that PSPC1-associated paraspeckle components including NONO, FUS, and the lncRNA *Neat1* are required for the PSPC1-induced IGF1R gene expression and cell motility. Our results provide lines of evidence that the PSPC1/IGF1R axis is the critical signaling to facilitate cancer cell motility and potentially as a therapeutic biomarker to predict malignant subtypes of cancer for receiving treatments with IGF1R inhibitor. 

IGF1R is a transmembrane tyrosine kinase receptor and considered as a potential therapeutic target due to associations of IGF1R overexpression with tumor metastasis, drug resistance, and poor prognosis in multiple cancer types [14,15,16,17]. Although numerous efforts were conducted for development of IGF1R inhibitors with promising anticancer effects in preclinical studies, high challenges of unable to prolong life of cancer patients were encountered in multiple clinical trials. Similar to other strategies for improving target therapy, one of the approaches is to identify biomarkers to select a suitable subgroup of cancer patients more susceptible to the treatment of IGF1R inhibitor. We speculate that PSPC1 overexpression in tumors could be a potential biomarker for suggestion of cancer patients to receive treatment of IGF1R inhibitor. First, our results demonstrated that PSPC1 is the upstream modulator of IGF1R expression to activate downstream FAK/Src focal adhesion signaling and facilitate cell motility that is crucial for tumor progression. Second, upregulation of PSPC1 and IGF1R as well as their downstream signaling are known to activate epithelial to mesenchymal transition (EMT), stemness, and metastasis in association with poor prognosis of cancer patients [11,42,43,44]. Finally, our results also demonstrated that treatments of IGF1R inhibitor PPP in HCC cells abolished PSPC1-activated IGF1R gene expression, downstream FAK/Src and AKT, as well as cell motility. 

We noticed that expression of PSPC1 in HCC cells induced morphological changes including cell spreading, cytoskeleton remodeling, polarized protrusion, and focal adhesion formation accompanied with modification in cell surface integrins for elevating cell motility. Based on the PSPC1-induced phenotypical transformation, we identified PSPC1 activated IGF1R/integrin signaling led to focal adhesion formation and cell surface interactions with integrin family proteins through integrated unbiased transcriptome and phospho-kinase array analysis. Our transcriptomic analysis showed that several integrin ligands were activated by PSPC1 including COL1A2, COL5A2, and LAMB1 and validated in PSPC1-expressing HCC cells (Figure 3). Consistently, activation of IGF1R was shown to promote cellular motility through altered cell surface integrin expression by activation of downstream IRS2, FAK, RHOA, and ROCK signaling pathways [25]. In fact, the collaborative signaling of IGF1R and integrins via stabilization of IGF1R/integrin cell adhesion complex was reported previously to promote cancer cell growth and motility in multiple cancer types [23,29,31,45]. Our results show that PSPC1 enhanced IGF1R expression and activated expression of several integrins, including integrins β1 and β4, coincided with previous studies. Since PSPC1 was shown to elevate matrix complex degradation enzymes matrix metalloproteinases (MMPs) for breaking the ECM physical barrier to facilitate metastasis [11], our detailed mechanism of PSPC1-activated dynamic ECM remodeling, cell surface integrin modulation along with stimulation of FAK/Src kinase signaling further advance our understanding PSPC1-driven tumor metastasis.

Compared to numerous studies of lncRNA *Neat1* participated in tumor progression in multiple cancer types [46,47], mechanistic studies for the roles of PSPC1 and the paraspeckle associated proteins in tumor progression is still in the infant stage. After proteomic and STRING networking analysis of PSPC1-interacting proteins, we prioritized NONO and FUS paraspeckle associated proteins along with lncRNA *Neat1* for evaluation of their roles in PSPC1-potentiated IGF1R activation. To our surprise, knockdown NONO, FUS, and lncRNA *Neat1* downregulated PSPC1-mediated IGF1R gene expression in multiple HCC cells. Our results suggested that NONO, FUS, and lncRNA *Neat1* are required for PSPC1/IGF1R axis enhanced cell motility. Although paraspeckle proteins are known to play either repressor or activator in gene transcription [1], our results might open up a new avenue for studying either paraspeckle or individual paraspeckle associated proteins participated in tumor progression and as the theranostic targets for developing innovative strategies for improving the life of cancer patients. Taken together, our findings provide a detailed mechanism for the PSPC1 activated IGF1R axis to enhance focal adhesion signaling and cell motility. Moreover, overexpression of PSPC1 or other paraspeckle proteins may be a potential biomarker for identification of subgroups of cancer patients for treatment with IGF1R inhibitor. 

## 5. Conclusions

Overall, our results suggest that PSPC1 controls cell adhesion and motility via induction of IGF1R expression. This induction leads to activation of integrins expression as well as FAK/Src and AKT signaling. PSPC1 RNA recognition motifs (RRM) are critical for IGF1R induction. In addition, PSPC1 collaborates with the paraspeckle core components FUS, NONO, and lncRNA *Neat1* to modulate IGF1R expression and facilitate cell motility (Figure 7).

## Figures and Tables

**Figure 1 cells-09-01490-f001:**
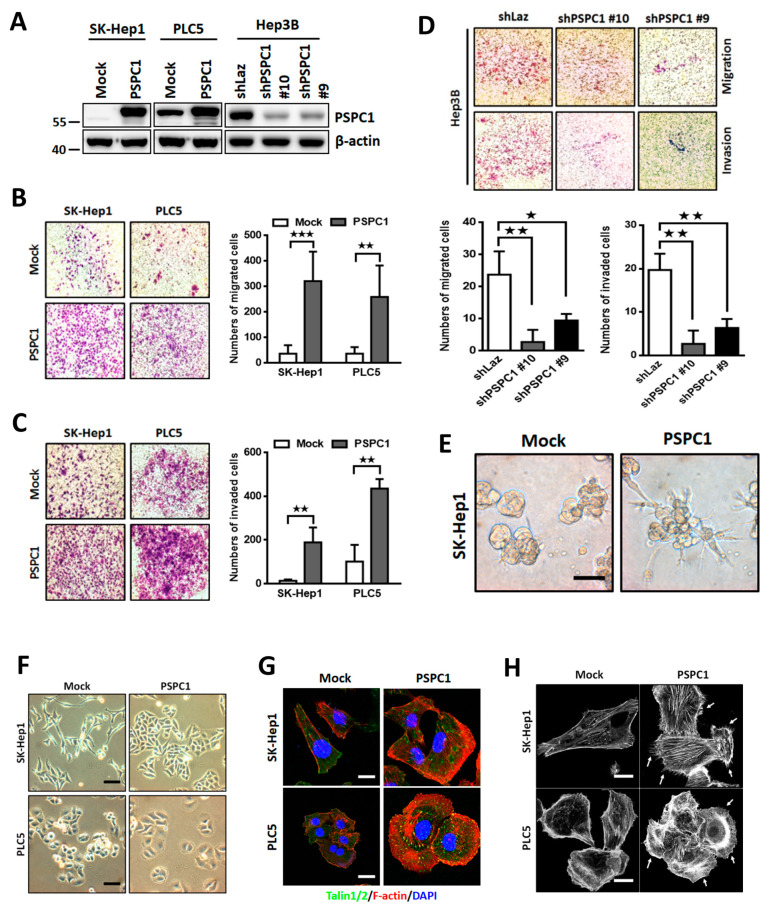
Overexpression of Paraspeckle protein 1 (PSPC1) promotes cancer cells migration, invasion, and actin cytoskeleton assembly that extend protrusions at cell leading edge. (**A**) PSPC1 protein levels in SK-Hep1, PLC5, and Hep3B cells with indicated treatments. (**B–D**) Cell migration and invasion capabilities were measured by Boyden chamber assay for 20 h with PSPC1 overexpressing or knockdown cells, and are summarized in the bar figures. The migrated cells stained with crystal violet dye are representative figures. (**E**) The capacity of cellular invasiveness was measured with PSPC1 overexpression group in compared to the control mock by using 3D-Matrigel coated plates for 6 days. Scale bar: 50 µm. (**F**) Phase-contrast image with morphological transformation of PSPC1 overexpressing HCC cells. Scale bars: 20 µm. (**G**) Co-distribution of f-actin and talin1/2 was detected by immunofluorescence staining in PSPC1 overexpressing cells compared to the mock control. Scale bars: 5 µm. (**H**) Phalloidin staining of SK-Hep1 and PLC5 cells expressing PSPC1 or mock control. White arrows point to the formation of f-actin filaments of filopodia or lamellipodia. Scale bars: 5 µm. Data are mean ± SD analyzed by paired and two-tailed *t*-test, *n* = 3 per group, *p*-values (* *p* < 0.05; ** *p* < 0.01; *** *p* < 0.001).

**Figure 2 cells-09-01490-f002:**
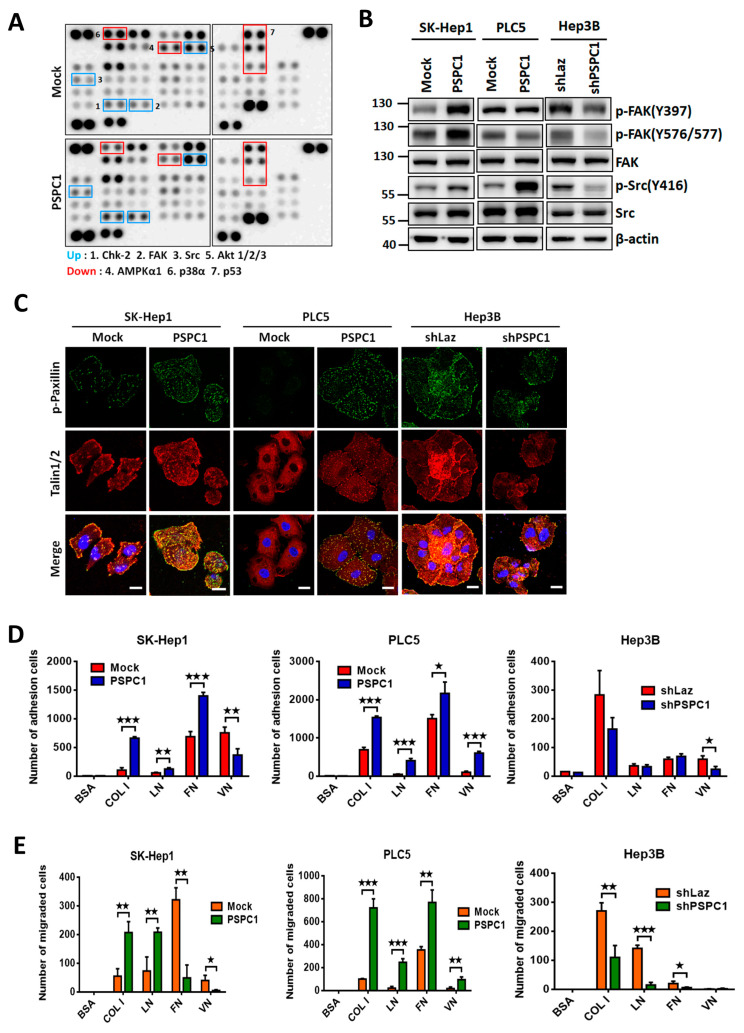
PSPC1 modulates FAK and Src kinases activity to promote cell adhesion and motility on extracellular matrix. (**A**) Phospho-kinase array was performed by using protein lysates of PSPC1-expressing SK-Hep1 cells. (**B**) Expression of phosphorylated FAK (Y397 and Y576/577) and Src (Y416) with their total proteins was investigated in three PSPC1-modulated HCC cells by Western blotting analysis. (**C**) Colocalization of phospho-Paxillin (green) and Talin1/2 (red) by double immunofluorescence staining in three HCC cells. Scale bars: 5 µm. (**D**) PSPC1-overexpressing and knockdown cells were seeded with serum free medium and plated into ECM coating cells of 96-well plate (COL I: collagen type I, LN: laminin, FN: fibronectin, VN: vitronectin) for 30 min. Adhered cells were counted in full field. (**E**) Capability of cell migration toward ECM were measured using Boyden chamber assay coated with ECM for 20 h in three PSPC1-modulating HCC cells. Data are shown in mean ± SD analyzed by paired and two-tailed *t*-test, *n* = 3 per group, *p*-values (* *p* < 0.05; ** *p* < 0.01; *** *p* < 0.001).

**Figure 3 cells-09-01490-f003:**
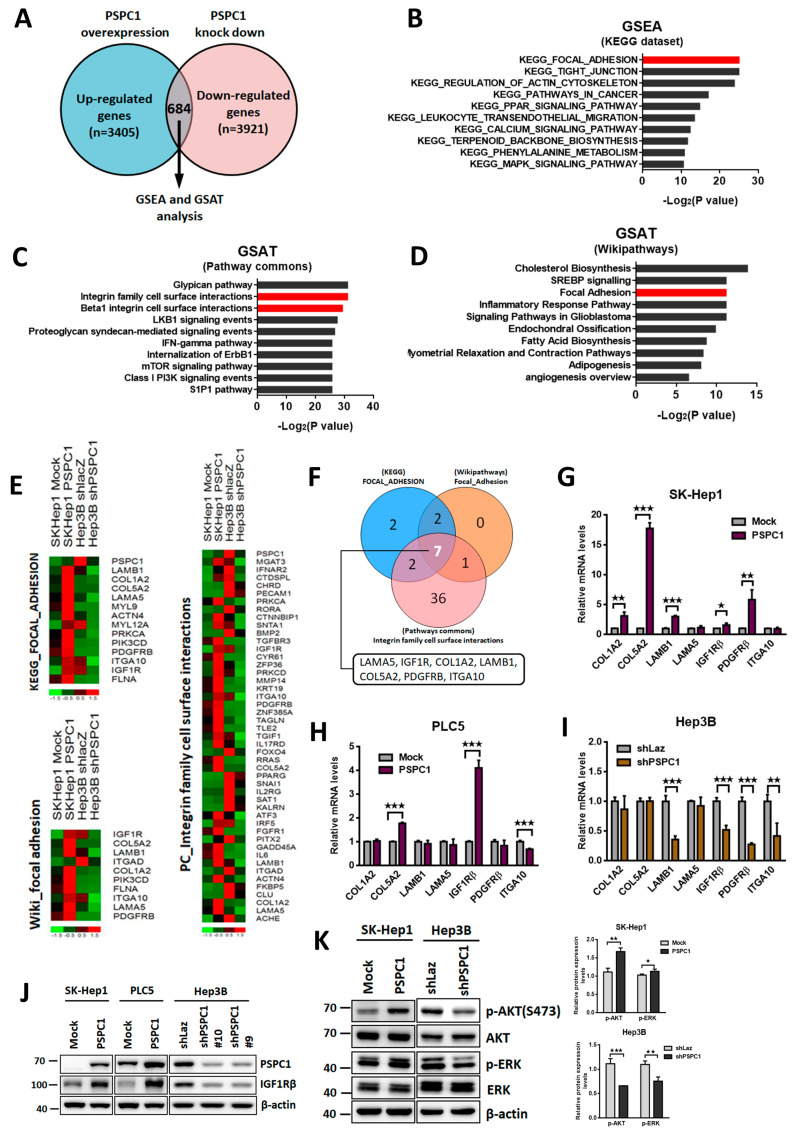
(**A**) Transcriptome analysis by RNA-Seq of next generation sequencing (NGS) was performed using PSPC1-expressing SK-Hep1 and PSPC1-depleted Hep3B cells. Intersection of upregulated genes and downregulated genes indicated a set of 684 genes responded to the PSPC1 modulation. (**B–D**) The top 10 enrichment pathways were displayed by histograms of KEGG, Pathway commons and Wikipathways data set by using GSEA and/or GSAT analysis. (**E**) PSPC1-modulated DEGs of the overlapping genes in the three independent enrichment pathways either upregulated (red) or downregulated (green) in PSPC1-expressing SK-Hep1 and PSPC1-depleted Hep3B cells as shown in heatmap diagrams. (**F**) Seven overlapped genes from three independent enrichment pathways are shown as a Venn diagram. (**G–I**) Reverse transcription quantitative PCR (RT-qPCR) analysis of seven PSPC1-modulated genes validated in three HCC cells. (**J**) IGF1R expression at protein level was measured by Western blotting assays in three PSPC1-overexpressing and knockdown HCC cells. (**K**) IGF1R canonical downstream signaling of p-AKT and p-ERK were detected by immunoblotting assay in PSPC1-expressing SK-Hep1 and PSPC1-depleted Hep3B cells. The right panels showed the quantification results of Western blotting analysis on the left. Data is shown in mean ± SD analyzed by paired and two-tailed *t*-test, *n* = 3 per group, *p*-values (* *p* < 0.05; ** *p* < 0.01; *** *p* < 0.001).

**Figure 4 cells-09-01490-f004:**
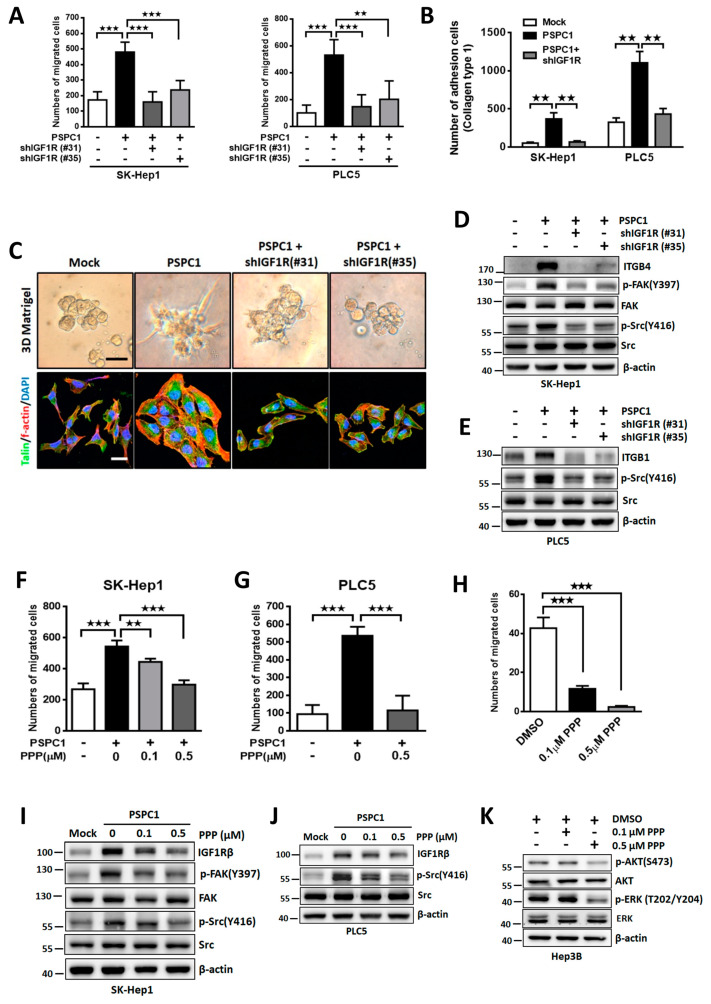
PSPC1 modulates cell adhesion and motility by activation of IGF1R signaling. Stable depletion of IGF1R by shRNAs in PSPC1-overexpressing SK-Hep1 and PLC5 cells reduced cell migration using Boyden chamber assay (**A**) and cell adhesion assay on type I collagen ECM (**B**). (**C**) PSPC1-induced invasiveness (top) and stress fiber assembly (bottom) were assessed by using 3D-Matrigel assay and f-actin staining, respectively, in PSPC1-overexpressing SK-Hep1 cells with and without depletion of IGF1R. (**D**,**E**) Knockdown of IGF1R suppressed PSPC1-activated p-FAK (Y397), p-Src (Y416), integrin β4, and integrin β1 in HCC cells detected by Western blotting assay. (**F**,**G**) Cell migration were detected by using Boyden chamber assay in PSPC1-overexpressing HCC cells treated with IGF1R inhibitor PPP in serum-free medium at indicated concentration for 20 h. (**H**) Cell migration ability was measured using Boyden chamber in parental Hep3B cells treated with PPP at indicated concentrations for 20h. (**I**,**J**) The expression of IGF1R, FAK, Src, phosphorylated-FAK, and phosphorylated-Src were detected in PSPC1-oveerexpressing SK-Hep1 and PLC5 cells treated with PPP in a dose dependent manner in serum-free medium for 24 h. (**K**) The expression of total AKT, ERK, p-AKT (S473), and p-ERK (T202/Y204) were detected in parental Hep3B cells treated with PPP in serum-free medium with the indicated concentrations for 24 h. Data are mean ± SD analyzed by paired and two-tailed *t*-test, *n* = 3 per group, *p*-values (** *p* < 0.01; *** *p* < 0.001).

**Figure 5 cells-09-01490-f005:**
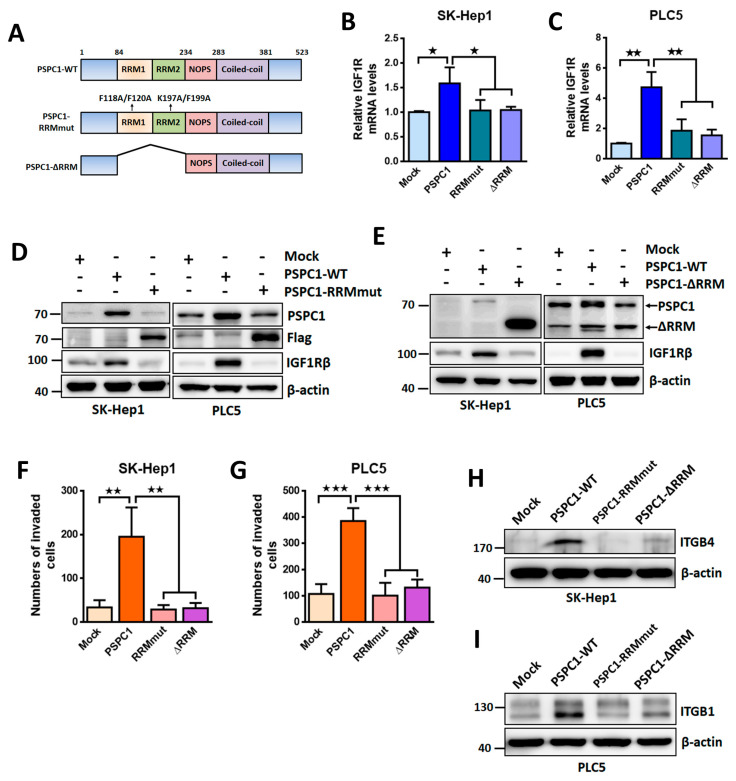
PSPC1 modulates IGF1R expression relying on PSPC1 RNA recognition domains (RRMs). (**A**) Diagram of the PSPC1 wild type (PSPC1-WT), PSPC1 RRM mutation at specific sites (PSPC1-RRMmut), and RRM deletion (PSPC1-ΔRRM) mutant. Expression of IGF1R at mRNA (**B**,**C**) and protein levels (**D,E**) detected by qRT-PCR and Western blot, respectively, in SK-Hep1 and PLC5 cells were compared in between expression of PSPC1-WT versus mutant of PSPC1-RRMmut or PSPC1-ΔRRM cultured for 2 days. (**F**,**G**) Cell invasion capability was detected in SK-Hep1 and PLC5 cells expressing PSPC1-RRMmut or PSPC1-ΔRRM in compared to PSPC1 wild type by Boyden chamber assay. (**H**,**I**) Integrin β4 and β1 protein levels were investigated by immunoblotting in SK-Hep1 and PLC5 cells expressing PSPC1-WT, PSPC1-RRMmut, and PSPC1-ΔRRM for 48 h. Data are mean ± SD analyzed by paired and two-tailed *t*-test, *n* = 3 per group, *p*-values (* *p* < 0.05; ** *p* < 0.01; *** *p* < 0.001).

**Figure 6 cells-09-01490-f006:**
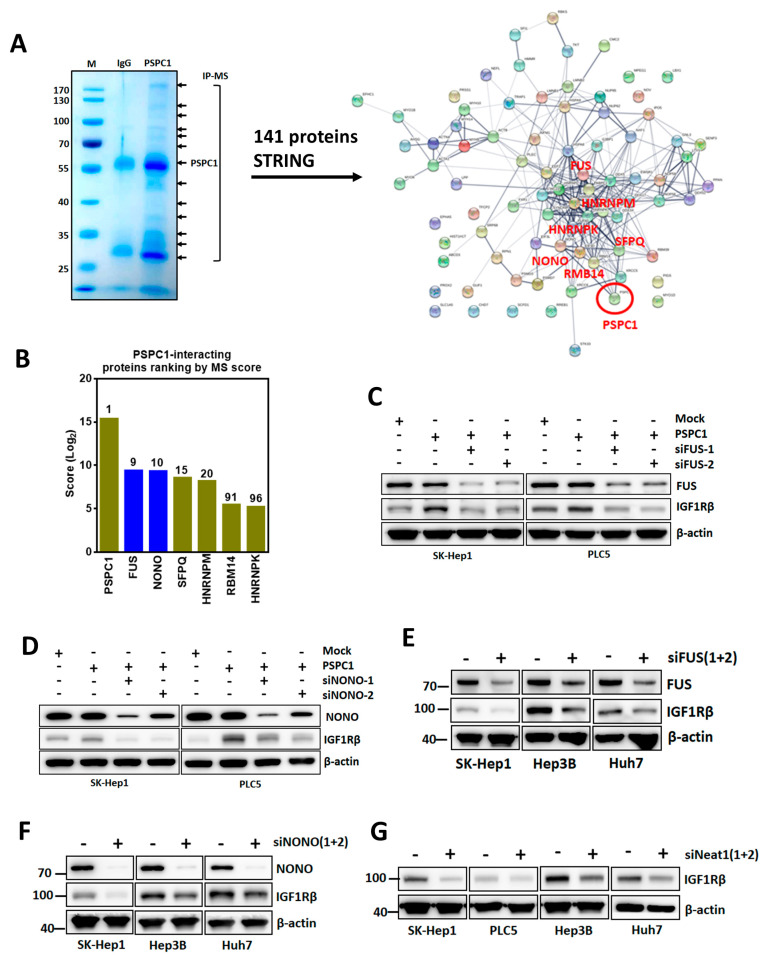
Paraspeckle core components PSPC1, FUS, NONO, and lncRNA Neat1 are essential for IGF1R induction. (**A**) Total cell lysates of Huh7 cells were immunoprecipitated using the PSPC1 antibody and then conducted SDS-PAGE for protein separation and stained with Coomassie Blue (left panel). Bands indicated by arrows were excised for in-gel digestion and LC-MS/MS analysis. A total of 141 interacting proteins were analyzed for protein–protein interaction networks by STRING (right panel). (**B**) The ranking list of top six PSPC1-interacting proteins in descending ordered by MASS score. (**C**,**D**) Depletion of FUS and NONO by siRNAs in PSPC1-overexpressing SK-Hep1 and PLC5 cells cultured for 2 days following performed immunoblotting assay for detection of IGF1R expression. (**E–G**) IGF1R expression diminished by immunoblotting analysis in knockdown of FUS, NONO, or *Neat1* with corresponding siRNAs in parental SK-Hep1, PLC5, Hep3B, and Huh7 cell lines cultured for 2 days.

**Figure 7 cells-09-01490-f007:**
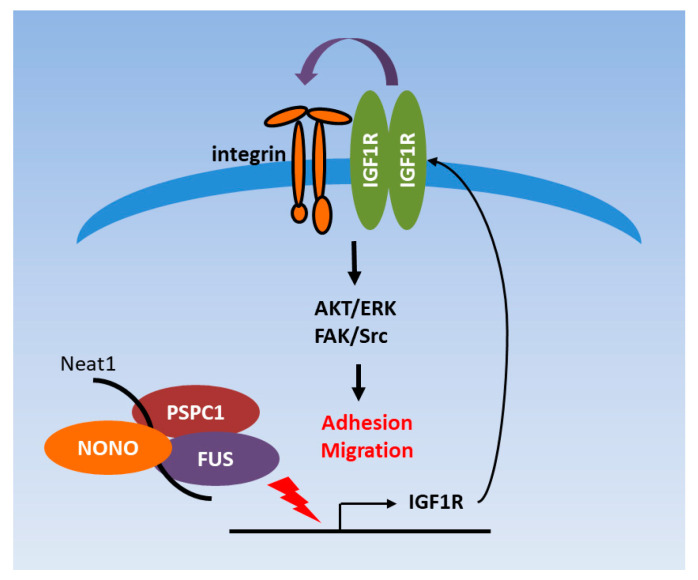
Hypothetical model of PSPC1/IGF1R axis to facilitate cell adhesion and motility.

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
