# Peer review of "PSPC1 Potentiates IGF1R Expression to Augment Cell Adhesion and Motility"

_cells, 2020, doi:10.3390/cells9061490_

Round 1
Reviewer 1 Report
The authors have provided lines of evidence that PSPC1 augments cell adhesion and motility via promoting IGF1R expression to stimulate downstream focal adhesion and integrin signaling pathways including integrin/FAK/Src and AKT axes, and they have investigated the impacts of paraspeckle component proteins and their binding scaffold lncRNA Neat1 participated in PSPC1/IGF1R axis-potentiated cell motility. The authors have performed transcriptome analysis by RNA-Seq of next generation sequencing (NGS) using PSPC1-expressing SK-Hep1 and PSPC1-depleted Hep3B cell, based on these data immunoblot analysis and several functional analyzes are used to clarify the PSPC1 controls cell adhesion and motility via induction of IGF1R expression. Furthermore, the authors have analyzed total of 141 interacting proteins for protein-protein interaction networks by STRING (right panel) and PSPC1-interacting proteins in descending ordered by MASS score.
FAK and IGF1R are also important for proliferative ability for tumor development, so please include any data or information on them concerning your experiments.
Author Response
Dear Reviewer
Please see the attachment file, thanks.

Reviewer 2 Report
The authors have assembled a manuscript making a case for the paraspecle protein 1 (PSPC1)
inducing the IGF1R protein to potentiate cell adhesion and motility. In general, the manuscript
is well assembled and contributes to making this case. However, there are a couple of weak
areas that should be addressed.
1. In the Materials and Methods the authors state that after transfecting cells with a PSPC1
containing vector they selected with neomycin, puromycin, and zeocine to get stable cell lines.
What is this vector and why does it need three concurrently added antibiotics to select? Also in
the figures the control cells are described as “mock” transfected. Is that an empty vector
control? Or untransfected cells…in which case are they also receiving the antibiotics?
2. The use of trysin-EDTA to resuspend cells for adhesion assays should result in loss of cell
surface and cell-cell adhesion receptors. Have you tried versene to resuspend the cells without
loss of surface proteins?
3. When describing the modified Boyden chambers using purified matrix proteins, the methods
indicate that you have coated the upper portion of the insert with the purified matrix proteins
and then added media with serum in the bottom of the well as the chemoattractant. The
description in the results section of the paper describe this as the cells migrating to the matrix
proteins. This is not correct. What you are looking at is the ability to migrate ON the matrix
protein to get to the pore to more toward the chemoattractant. The interpretation of the
results is then different. Does one matrix molecule allow the cells to migrate more easily to the
pore and the attractant gradient than another?
4. For the invasion and migration assays, the methods say that the cells on the underside of the
filter were counted. Was that all the cells? A number of random fields?
5. The methods section numbered 2.8 is a bit hard to follow. This is not the only place where
you could use a bit more help with the English in the writing.
6. The results from the quantification of your adhesion and migration/invasion experiments are
quantitative. However, the authors also want to a point of changes in cell morphology, stress
fibers, and focal adhesion contacts without any quantification to back this up. These are all
parameters for which there are accepted quantification methods in cell biology. In figure 1 it
looks like over-expression of PSPC1 results in cells that grow more as if they are forming
colonies. They appear to be maximizing cell-cell contacts like you would see with addition of a
cadherin. These phenotypic changes need quantifying.
7. In figure 2B, the increase in pFAK in the PLCS line is not convincing. You should consider
quantitation from multiple blots.
8. In figure 2C, the morphological changes in the cells should be quantified/characterized.
Certainly, in the presence of PSPC1 they are not switching to a mesenchymal phenotype that
would be more consistent with increased migration or invasion.
9. A better and more complete interpretation is needed for the data in figure 2.
10. Does addition or repression of PSPC1 alter cell proliferation?
11. For line 263, I do not understand what the authors are trying to get across. Please clarify.
12. While the first parts of Figure 3 provide interesting information that provides the basis for
the further analysis of the role of IGF1R, part K is not terribly convincing. Quantification might
help tell us if these changes are meaningful and reproducibly.
13. For Figure 4, there should be a Western blot to demonstrate that altering the IGF1R levels
does not change the PSPC1 levels to go along with the migration data. Same with supplemental
figure 2.
14. What is the control vector for IGF1R? When ISG1R is added to cells with KD of PSPC!, what
is the result for the vector without insert?
15. This manuscript contains a lot of data. So much so that from Figure 5 forward, there is an
almost overwhelming amount of information. To help the reader follow this, the authors
should expand the results sections for these figures to more fully develop their story and lead
the reader through all the data and its significance.
Author Response
Please see the attachment, thanks.
Best regards,
